# Molecular Fingerprint of Endocannabinoid Signaling in the Developing Paraventricular Nucleus of the Hypothalamus as Revealed by Single-Cell RNA-Seq and In Situ Hybridization

**DOI:** 10.3390/cells14110788

**Published:** 2025-05-27

**Authors:** Evgenii O. Tretiakov, Zsófia Hevesi, Csenge Böröczky, Alán Alpár, Tibor Harkany, Erik Keimpema

**Affiliations:** 1Department of Molecular Neurosciences, Center for Brain Research, Medical University of Vienna, 1090 Vienna, Austria; evgenii.tretiakov@meduniwien.ac.at (E.O.T.); zsofia.hevesi@meduniwien.ac.at (Z.H.); csenge.boeroeczky@meduniwien.ac.at (C.B.); 2Department of Anatomy, Histology and Embryology, Semmelweis University, 1085 Budapest, Hungary; alpar.alan@semmelweis.hu; 3SE NAP Research Group of Experimental Neuroanatomy and Developmental Biology, Semmelweis University, 1085 Budapest, Hungary; 4Department of Neuroscience, Biomedicum 7D, Karolinska Institutet, 17165 Solna, Sweden

**Keywords:** cannabinoid receptor, endocannabinoid, hypothalamus, monoacylglycerol lipase, postnatal development, *sn*-1-diacylglycerol lipase

## Abstract

The paraventricular nucleus of the hypothalamus (PVN) regulates, among others, the stress response, sexual behavior, and energy metabolism through its magnocellular and parvocellular neurosecretory cells. Within the PVN, ensemble coordination occurs through the many long-range synaptic afferents, whose activity in time relies on retrograde neuromodulation by, e.g., endocannabinoids. However, the nanoarchitecture of endocannabinoid signaling in the PVN, especially during neuronal development, remains undescribed. By using single-cell RNA sequencing, in situ hybridization, and immunohistochemistry during fetal and postnatal development in mice, we present a spatiotemporal map of both the 2-arachidonoylglycerol (2-AG) and anandamide (AEA) signaling cassettes, with a focus on receptors and metabolic enzymes, in both molecularly defined neurons and astrocytes. We find type 1 cannabinoid receptors (*Cnr1*), but neither *Cnr2* nor *Gpr55*, expressed in neurons of the PVN. *Dagla* and *Daglb*, which encode the enzymes synthesizing 2-AG, were found in all neuronal subtypes of the PVN, with a developmental switch from *Daglb* to *Dagla*. *Mgll*, which encodes an enzyme degrading 2-AG, was only found sporadically. *Napepld* and *Faah*, encoding enzymes that synthesize and degrade AEA, respectively, were sparsely expressed in neurons throughout development. Notably, astrocytes expressed *Mgll* and both *Dagl* isoforms. In contrast, mRNA for any of the three major cannabinoid-receptor subtypes could not be detected. Immunohistochemistry validated mRNA expression and suggested that endocannabinoid signaling is configured to modulate the activity of afferent inputs, rather than local neurocircuits, in the PVN.

## 1. Introduction

The hypothalamus is an archetypical, evolutionary-conserved vertebrate brain structure that controls essential physiological processes, including food and fluid intake, energy homeostasis, stress responses, sleep/wake cycles, as well as sexual behavior and reproduction, to ensure the subject’s survival [1]. Consequently, the hypothalamus contains a collection of topographically segregated neuronal loci, each possessing their own molecular signature and synaptic connectivity. For instance, the paraventricular nucleus (PVN), flanking the third ventricle bilaterally and mainly consisting of excitatory vesicular glutamate transporter 2 (*Slc17a6*/VGLUT2)^+^ neurons, integrates neuroendocrine and autonomic functions through its dorsolateral magnocellular and medioventral parvocellular subdivisions. Its neurosecretory magnocellular cells are large-bodied neurons expressing either vasopressin (m^AVP^) or oxytocin (m^OXT^), with both subpopulations projecting directly to the posterior pituitary where they release their hormones directly into the systemic circulation to regulate blood pressure, water-balance, and maternal, social and feeding behaviors, respectively [2]. In contrast, parvocellular (small-bodied) neurosecretory neurons contain corticotrophin-releasing hormone (CRH; p^CRH^), thyrotropin-releasing hormone (TRH; p^TRH^), or somatostatin (SST; p^SST^) and extend their axons to the median eminence, a gateway structure for hypothalamic hormones to reach the anterior pituitary through the hypophyseal portal system. Once released into the portal blood, CRH orchestrates the hypothalamus–pituitary–adrenal (HPA) axis upon stress, with sequential release events for adrenocorticotropic hormone (ACTH) from the pituitary and cortisol from the adrenal cortex. TRH stimulates the release of thyroid-stimulating hormone (TSH) from the anterior pituitary, thus controlling metabolic and cardiovascular function through the hypothalamus–pituitary–thyroid (HPT) axis. Antagonistically, SST inhibits the release of TSH from the anterior pituitary [3]. Thus, the PVN is essential for bodily metabolism, with its dysfunction implicated in obesity, hypertension, diabetes insipidus [2], as well as anxiety and depression [4,5].

To control the many physiological processes, the PVN is innervated by long-distance afferents, both excitatory and inhibitory in nature, originating from multiple brain regions, e.g., from excitatory proopiomelanocortin (POMC)^+^/*Slc17a6*^+^ glutamatergic [6], and inhibitory agouti-related protein (AgRP)/neuropeptide Y neurons [7] of the arcuate nucleus to control energy expenditure [8,9]; glutamatergic neurons of the amygdalo–piriform transition area, and inhibitory neurons of the central amygdala to modulate CRH secretion upon fear induction [10]; as well as inhibitory neurons of the bed nucleus of the stria terminals upon reward seeking and stress [11]. As a general principle, the temporal precision of both the innate neurocircuits and neuroendocrine output of the PVN is controlled by inhibitory neurotransmissions originating in nearby brain areas, rather than intrinsically, through GABA_A_ receptor-dependent inhibition [12,13,14]. However, synaptic integration is unlikely to be exclusively reliant on distant-positioned neurons [12,13], particularly in the absence of a major interneuron (GABA) contingent locally (Figure 1a_1_,b) [15]. Thus, we hypothesized that synaptic synchronization and local neurotransmitter availability, from both local collaterals and afferents, could additionally rely on fast-acting retrograde signaling. Indeed, compared to other hypothalamic areas, the PVN is known for its expression of type 1 cannabinoid-receptor mRNA (*Cnr1*/CB_1_R) [16], a G_i/o_ protein-coupled receptor negatively affecting neurotransmitter release (Figure 1a_2_,b–b_2’_). The activation of presynaptic CB_1_Rs occurs through 2-arachidonoylglycerol (2-AG) [17] and anandamide (AEA) [18], endocannabinoids produced in subsynaptic dendrites [19]. Indeed, engagement of CB_1_Rs in the PVN was shown to influence energy metabolism [20,21], food intake [20,22], and the stress response [23,24]. However, the cellular architecture of the endocannabinoid system on identified neuronal and astrocyte populations in the PVN remains unresolved, whether during development or in adulthood.

Here, we produced a comprehensive expression map of cannabinoid receptors (*Cnr1*/*Cnr2*/*Gpr55*), as well as the putative enzymes involved in 2-AG (*Dagla*/*Daglb* vs. *Mgll*) and AEA (*Napepld*/*Gde1* vs. *Faah*) synthesis and degradation at pre- and postnatal stages of the developing mouse PVN. Particularly, by using in situ hybridization, we found that m^OXT^ and m^AVP^, which appear at embryonic day (E) 15.5, expressed *Cnr1*, but neither *Cnr2* nor *Gpr55* mRNA until postnatal day (P) 21 (pre-adolescence). Once the PVN became enriched in p^TRH^/p^CRH^ neurons, these neuronal subtypes also expressed *Cnr1*. Spatiotemporal *Dagla*/*Daglb* and *Mgll* expression tightly followed that of *Cnr1*, also recapitulating a robust *Daglb-to-Dagla* switch at neonatal life, in line with the predominance of DAGLα in retrograde neurotransmission at mature synapses [25]. For p^SST^ neurons, which segregate last in the PVN, molecular arrangements were indistinguishable from other neuronal subtypes, yet expressional patterns were delayed as much as the emergence of p^SST^ neurons themselves. *Napepld* mRNA content was minimal in all neurons, while *Faah* expression increased towards P21. Immunohistochemistry confirmed protein expression sites, with DAGLα apposing CB_1_R labeling. Single-cell RNA-seq data showed a near complete lack of cannabinoid receptors in astrocytes of the adult PVN, but revealed *Dagla* expression. We found reliable amounts of neither *Napepld* nor *Faah* mRNA in astrocytes, reflecting previous data suggesting that astrocytes are more reliant on 2-AG signaling when interacting with nearby neurons [26,27]. In sum, we described the molecular constituents of the endocannabinoid system in identified magnocellular and parvocellular neurons, as well as astrocytes, at single-cell-precision during fetal and postnatal development of the PVN.

## 2. Results

**Neuronal diversity in the PVN during brain development.** We first examined the expression of *Cnr1* in relation to *Slc17a6* (VGLUT2) and *Slc32a1* (VGAT) in the adult PVN (Figure 1a,a_1_). We note that *Slc32a1* was sparse in the adult PVN, particularly against its immediate surroundings, suggesting that local (inhibitory, GABAergic) neurocircuits therein are uncommon (Figure 1a_1_,b,b_1_). In contrast, *Slc17a6* mRNA expression was abundant. *Cnr1* expression dominated in *Slc17a6*^+^ neurons, but also co-localized with a small pool of *Slc32a1*-containing cells (Figure 1a_2_,b–b_2’_). By using whole-brain spatial in situ hybridization for *Cnr1* at postnatal day 7, we confirm *Cnr1* expression at earlier developmental time stages throughout the entire anterior–posterior axis, with labeling of the PVN standing out in the hypothalamic area (Figure 1c,c_1_).

Next, we defined neuronal subtypes in the PVN by combining neurotransmitter- and neuropeptide-related gene signatures in embryonic (E15.5–E17.5), neonatal (P0), juvenile (P2, P10), and pre-adolescent (P23) hypothalamus datasets. mRNA expression levels were visualized through uniform manifold approximation and projection (UMAP) plots in two dimensions to distinguish neuronal subtypes (Appendix A) [15]. At E15.5, magnocellular neurosecretory cells expressing *Oxt* (m^OXT^) and *Avp* (m^AVP^) mRNA appeared first, with mRNA levels gradually rising to P21 (Figure 2a–a_1_). We caution that the sequence similarity between *Oxt* and *Avp* could have limited the precise assessment of *Avp* mRNA present, and for this reason *Avp* was excluded from subsequent analysis [28]. Next, we distinguished parvocellular cells at E17.5, particularly p^CRH^ and p^TRH^ neurons, as well as a late-emerging p^SST^ group after birth (Figure 2a_2_–a_4_). For parvocellular cells, mRNA levels increased as a factor of developmental stage and peaked at P10-to-P21. The temporal pattern described here was validated by an RNA-sequencing dataset published earlier [29], which also show m^OXT^/m^AVP^ > p^CRH^/p^TRH^ > p^SST^ neurons, but at a shallower resolution (Appendix A). This sequence of events is compatible with the concept that long-range projection neurons of the hypothalamus mature earlier than locally targeting neurons and/or interneurons [15].

**Assembly of the endocannabinoid system.** To map genes broadly associated with the ‘endocannabinoid system’, we first examined gene expression for *Cnr1*, *Cnr2* [16,30], and *Gpr55* [31] and visualized the intersection of gene expression sets with *UpSet* plots. We found *Cnr1* expression at molecule numbers exceeding 3000 per cell at E15.5 and E17.5 in subsets of m^OXT^, p^TRH^, and p^CRH^ neurons, with its levels maintained until P21 (>1000 mRNA copies per cell; Figure 3a,b). In p^SST^ neurons, *Cnr1* was detected in 15–20% of cells only postnatally, correlating the temporal expression pattern of *Sst* itself (Figure 2a_4_). Considering the small sample size of p^SST^ neurons, their molecular features were not processed further. Even though *Cnr2* mRNA was previously found in the hypothalamus [32], we could not reliably detect *Cnr2* mRNA transcripts by single-cell RNA-seq in neurons, suggesting limited, if any, *Cnr2* contributions to neuronal development, at least in the PVN (Figure 3a_1_). In contrast, we detected a few *Gpr55*-containing neurons, mainly m^OXT^ and p^TRH^ cells, with low mRNA copy numbers (<100 mRNA copies per cell) at neonatal and juvenile ages (Figure 3a_2_,b) [33]. Thus, *Cnr1* is the main cannabinoid receptor in neurons of the PVN to respond to (endo-)cannabinoid signals.

Next, we assessed *Dagla*, *Daglb* [34], and *Mgll* [35] mRNA expression. Both *Dagla* and *Daglb* were found in m^OXT^, p^TRH^, and p^CRH^ neurons at both E15.5 and E17.5, with mRNA levels, but not cell abundance, diminishing into pre-adolescence (Figure 4a,a_1_,b). Expression of *Dagla*/*Daglb* in p^SST^ cells was delayed to postnatal life as much as was seen for *Cnr1* and *Sst* mRNAs (Figure 2a_4_ and Figure 4a). During embryonic stages, we found a ~2.5-fold higher number of neurons expressing *Daglb* compared to *Dagla* (84 vs. 35 cells, over m^OXT^ and p^TRH^ neurons), which reversed between E17.5 and P10 (52 vs. 86 cells; Figure 4b) corroborating a developmental isoform switch model proposed earlier [25,34,36]. In parallel, *Mgll* expression gradually decreased towards P21. Of note, *Mgll* mRNA was found in the same neuronal clusters, but not cells, as *Dagla*/*Daglb* (Figure 4a_2_,b). These data indicate that molecular and rate-limiting constituents of 2-AG signaling are present throughout neuronal diversification in the PVN.

Subsequently, we determined the cellular foci for *Napepld* mRNA expression, which is implicated in AEA synthesis [37,38]. *Napepld* mRNA was sparse in both magnocellular and parvocellular PVN neurons at any time point (Figure 5a,b). In contrast, *Gde1* expression, contributing, among a plethora of other metabolic processes, to AEA synthesis [39], was promiscuous and significant in all neuronal clusters (Figure 5a_1_,b). The expression of *Faah*, the enzyme chiefly degrading AEA [40], was pronounced (>500 mRNA molecules/cell) throughout brain development in both magnocellular and parvocellular neurons (Figure 5a_2_,b).

**Nanoscale anatomy of 2-AG signaling revealed by in situ hybridization.** We have performed multiplexed fluorescence in situ hybridization to reconstruct 2-AG signaling (Figure 6). At E15.5, the prospective territory of the PVN was outlined by *Cnr1* mRNA expression, which was reminiscent of the shape of the adult structure (Figure 6a,a_1_ vs. Figure 1b). *Cnr1* mRNA localized to the majority of p^CRH^ neurons (Figure 6a_1,_a_2_ and Appendix A), which is similar to subpopulations of m^OXT^ and p^TRH^ neurons (Figure 6a_3,_a_4_). While *Daglb* mRNA transcripts were found in most p^CRH^, m^OXT^, and p^TRH^ cells (Figure 6a_1_–a_4_), *Dagla* was not detected reliably by in situ hybridization at E15.5, even if we excluded technical frailties by visualizing *Dagla* in the hippocampus *en masse* during embryogenesis (Appendix A). Thus, we continued with the analysis of *Daglb* for developmental stages and found *Daglb* mRNA abundantly at E15.5 (Figure 6a_2_ and Appendix A). At P10, both *Cnr1* and *Daglb* remained expressed in both p^TRH^ and m^OXT^ cells (Figure 6b–c_1_). *Mgll* was present in m^OXT^ neurons and co-existed with *Cnr1* (Figure 6c_1_). At P21, *Daglb* and *Mgll* were rarely expressed. In contrast, *Dagla* co-localized with *Cnr1* primarily in p^TRH^ neurons (Figure 6d–d_2_). Similarly, we continued to detect *Cnr1* mRNA in p^CRH^ cells (Figure 6e,e_1_) at P21.

Subsequently, we performed immunohistochemistry for DAGLα and CB_1_Rs at E15.5 and P21. In accord with our results using in situ hybridization, CB_1_R protein accumulated in the fetal PVN, and decorated processes, which can be interpreted as labeling of growth cones and nascent synapses on afferent inputs. In contrast, DAGLα immunoreactivity was minimal (Figure 6f,f_1_). At P21, CB_1_Rs continued to decorate terminal-like punctae, encircling perikarya and processes of, e.g., m^OXT^-containing neurons (Figure 6g,g_1_), with DAGLα confined mostly along the dendrites of neurons and in apposition to CB_1_Rs (Figure 6g,g_2_). In sum, cell-resolved neuroanatomy confirmed the cell identities, transcript switches for *Dagla* and *Daglb*, the developmental dynamics for *Cnr1* mRNA, and the classical configuration of DAGLα and CB_1_R for retrograde signaling at the protein level.

**The endocannabinoid system in astrocytes.** As endocannabinoid signaling is essential for neuron-to-astrocyte communication in multiple physiological processes [41,42,43], we addressed if cannabinoid receptors and enzymes related to endocannabinoid metabolism were expressed in astrocytes in the pre-adolescent PVN. We justify the choice of P21 by the postnatal window of astrocytogenesis and maturation that sequentially occur between P2-P14 [44]. Thus, terminally differentiated astrocytes that could reflect bona fide PVN-related signaling could be resolved starting at P21 [44]. We subdivided astrocytes based on their prototypical markers into excitatory amino acid transporter 1 (*Slc1a3*) [45], glial fibrillary acidic protein (*Gfap*) [46], and aldehyde dehydrogenase 1 family member L1 (*Aldh1l1*) [47], which jointly defined a cellular cluster (Figure 7a). When plotting cannabinoid receptors, we found a near complete lack of *Cnr1*, *Cnr2*, and *Gpr55* mRNA (Figure 7b,c), which was unexpected as functional CB_1_Rs have previously been localized to hypothalamic nuclei other than the PVN [48], as well as also to extrahypothalamic regions [26,42,49,50]. Both *Dagla* and *Daglb* mRNA were found in subsets of astrocytes, while *Mgll* mRNA was present in a larger subset (Figure 7b_1_,c). Despite *Gde1* being present in astrocytes, neither *Napepld* nor *Faah* could be detected reliably (Figure 7b_2_,c). We validated the above data by in situ hybridization, which showed the lack of *Cnr1* mRNAs but the presence of *Dagla* (Figure 7d,d_1_). Thus, astrocytes in the PVN could contribute to endocannabinoid metabolism and assist in shaping synaptic neurotransmission [51].

## 3. Discussion

This study describes the developmental dynamics of cannabinoid receptor- and endocannabinoid metabolism-related enzyme expression in the PVN of mice. Major findings include that all neurons go through a CB_1_R^+^ phase during embryonic and postnatal development, which is compatible with previous proposals of CB_1_R being ubiquitously expressed upon neurogenic commitment in vertebrates [52,53] and is associated with axonal growth and guidance. For 2-AG synthesis, our data recapitulates a developmentally regulated *Daglb*-to-*Dagla* switch that has been shown for the cerebral cortex [54] but not before for the hypothalamus. Moreover, the finding that *Daglb* and *Dagla*, but lower amounts of *Mgll*, are expressed in neurons of the PVN is compatible with known anatomical arrangements, with the PVN being reliant on monosynaptic afferents from external sources (bed nucleus of the stria terminalis, amygdala, arcuate nucleus, and suprachiasmatic nucleus) rather than local interneurons to synchronize its endocrine output. Accordingly, MAGL, together with CB_1_Rs, are expected to be expressed outside the PVN to time the presynaptic action of endocannabinoids produced by PVN neurons. Thus, the spatial configuration of endocannabinoid signaling during neuronal diversification in the PVN and in adulthood is adequate to control the avalanche of synaptic inputs arriving from intra- and extrahypothalamic areas. In addition, the importance of retrograde neurotransmission in the PVN is highlighted by the use of gaseous neuromodulators, particularly nitric oxide (NO). Neuronal NO synthase (*Nos1*) is particularly abundant in the PVN as compared to other hypothalamic nuclei, and its effects on food intake [55,56], renal sympathetic nerve activity [57,58], and sympatho-adrenomedullary outflow [59] are well described.

Besides cannabinoid receptors, we mapped the cellular distribution of *Dagla*, *Daglb*, *Napepld*, *Gde1*, and *Faah*, enzymes that are fundamental to the turnover of endocannabinoids. Even though these enzymes are the most abundant to modulate endocannabinoid levels in the brain and at the periphery, we acknowledge the contributions of others that have not been studied, e.g., MGL degrades ~85% of 2-AG in the brain [60]. In contrast, α/β-hydrolase domain containing 6 (*Abhd6*) [61] and 12 (*Abhd12*) [62] account for ~4% and ~9% of 2-AG degradation, respectively [60]. Similarly, the metabolic pathway for AEA is complex and includes *Abhd4*, *Gde4/7*, and *Ptpn22*, as well as *Lox* and *Cox2*, for synthesis and degradation, respectively (for extensive reviews see Refs. [63,64]). As the biological significance of these enzymes is less understood, we did not pursue them in this report. We also note that *Gde1* has been found important to AEA synthesis, at least in vitro. Yet, AEA levels in the brain of *Gde1*^−/−^ mice do not differ, questioning its role in AEA synthesis in vivo [65]. Therefore, and given the low abundance of *Napepld* mRNA expression, we hypothesize that 2-AG signaling might be more prevalent than AEA in the PVN. While we are confident that our study still provides substantial insights into the architecture of endocannabinoid signaling in the PVN, our datasets will certainly be amenable for further focused analysis.

2-AG is considered to be the main circuit-breaker driving retrograde signaling in a fast, but phasic, manner [66,67], since 2-AG is a more potent agonist at the CB_1_R [68] and is available at up to 1000-fold higher concentrations in the brain [18] (but see Ref. [66] for methodological caveats), and its absence curtails retrograde signaling in the DAGLα knockout mice [25]. Conversely, AEA is regarded as tonically inhibiting neuronal activity [67,69]. In accord with this hypothesis, previous reports suggest that tonic regulation of the HPA axis by AEA occurs either directly at neurons of the PVN [70] or indirectly through extrahypothalamic sites, such as the basolateral amygdala [69]. Nevertheless, mRNA levels predict neither protein abundance nor enzymatic activity [71]. Therefore, cell-resolved electrophysiology will be best placed to define the contribution of these endocannabinoids to regulating specific behaviors.

While endocannabinoids limit neurotransmitter release through neuronal CB_1_Rs, recent studies have proposed that CB_1_Rs on astrocytes could instead promote neuronal activity by controlling metabolite availability [27]. Accordingly, activation of mitochondrial CB_1_Rs in astrocytes could regulate glucose metabolism to ensure neuronal bioenergetics [49]. Furthermore, CB_1_Rs localized to astrocyte leaflets ensheathing blood vessels in the nucleus accumbens were implicated in regulating anxiety and depression-like behaviors [50]. In the hypothalamus, engagement of CB_1_Rs on or in astrocytes might regulate processes involved with energy metabolism, including leptin signaling and glycogen storage [48]. Therefore, we expected astrocytes in the PVN to express CB_1_Rs (and probably also other cannabinoid receptors). Nevertheless, we could not detect any with the tools available this time. We did however find substantial expression of both DAGL isoforms, implicating astrocytes in 2-AG metabolism. As 2-AG can be released by astrocytes in response to other non-endocannabinoid-mediated signals, such as ATP [72], we propose that astrocytes could tune neuronal activity through 2-AG release. Thus, our findings suggest novel, metabolism-driven endocannabinoid availability as a potential rate-limiting step for the processing of synaptic inputs and translating those into hormonal output at the level of PVN neurons.

## 4. Materials and Methods

**Ethical considerations.** Mice were housed under standard husbandry conditions in a temperature- and humidity-controlled room (12 h/12 h light cycle, 55% humidity, and 22–24 °C ambient temperature). Animals had ad libitum access to food and water throughout. The maintenance and welfare of the animals conformed to the 2010/63/European Communities Council directive. Experimental protocols for tissue collection were approved by the Austrian Ministry of Science and Research (66.009/0145-WF/II/3b/2014 and 66.009/0277-WF/V3b/2017). Pregnant (embryonic day (E) 15.5), and postnatal day (P) 4, P10, and P21 C57BL/6JRj mice were obtained from Janvier Labs and kept on site as adequate.

**Single-Cell RNA-seq data acquisition, and harmonization.** This study utilized two publicly available single-cell RNA-seq datasets to examine the expressional dynamics of genes related to endocannabinoid signaling during mouse hypothalamus development. The principal dataset comprised 51,199 cells, 24,340 features, and encompassed all developmental stages [15]. A second dataset was obtained as a pre-processed AnnData object, and contained 128,006 cells profiled for 27,998 genes along complementary developmental time points [29,73]. All subsequent analyses were performed in *R* (version 4.3 or higher) within a reproducible workflow framework documented using Quarto notebooks [74]. Gene expression matrices were appropriately transposed, and cell/gene metadata were standardized. Uniform manifold approximation and projection (UMAP) coordinates within the dataset were integrated as distinct dimensionality reduction objects within Seurat [75,76,77]. Metadata pertaining to developmental stages and original cluster annotations were preserved. The Romanov et al. [15] dataset, available as a Seurat RDS file, was loaded and updated to the latest Seurat object specifications. Cluster identities and associated color palettes were harmonized across both datasets to facilitate comparative analyses.

**Cellular quality control and filtering.** Stringent quality control was performed on both datasets to exclude potentially compromised cells and technical artifacts. Cells were retained if they had more than 500 unique genes and fewer than 25,000 unique molecular identifiers (UMIs). Additional filtering excluded cells based on high percentages of mitochondrial, ribosomal, or hemoglobin genes, elevated doublet prediction scores, and low transcriptomic complexity (log10 genes per UMI). These thresholds were enforced to enrich high-quality cellular profiles while acknowledging the potential exclusion of certain biological states, as per any filtering strategy [15,78].

**Gene set curation and expression analysis.** Gene sets representing canonical endocannabinoid receptors (*Cnr1*, *Cnr2*, and *Gpr55*), metabolic enzymes (*Dagla*, *Daglb*, *Gde1*, *Mgll*, *Napepld*, and *Faah*), and neuropeptides pertinent to hypothalamic classification (*Oxt*, *Avp*, *Crh*, *Sst*, and *Trh*) were defined based on established literature [19,79,80,81,82]. To focus analyses on more robustly expressed genes and to mitigate noise from sparse detection, expression matrices were filtered to include only genes ranking above the 40th percentile of mean expression across all cells. For qualitative assessment of co-expression, gene expression was binarized based on detection status (expression > 0.5th percentile for each gene separately) within individual cells. Therefore, intersectional co-expression patterns among key gene sets could be visualized using *UpSet* plots generated with the *UpSetR* package [78,83]. UMAP was used to visualize high-dimensional cellular states in two dimensions, primarily using coordinates provided in the original datasets or recomputed as necessary [15,29,76,77,78]. The expression patterns of specific genes and the aggregate expression of curated gene sets were visualized across developmental stages and cell clusters using Seurat’s *FeaturePlot* function, optimizing parameters such as point size and transparency for clarity [75].

**Descriptive statistics and exploratory analysis.** Quantitative summaries of gene expression, including means, standard deviations, and quartiles, were calculated for the specified target genes at the developmental periods indicated. The *skimr* package was used for descriptive statistics for numerical metadata and expression features, ensuring transparency regarding data distributions and completeness [78].

**Brain clearing and light sheet microscopy.** Whole postnatal day 7 brains were cleared and processed for in situ hybridization according to published protocols [84], with the exception that we used 30 pmol/mL, instead of 60 pmol/mL, of the fluorescent hairpin. All samples were imaged in dibenzyl ether with a measured refractory index of 1.55. Images were acquired on a Lightsheet 7 microscope (Zeiss, Jena, Germany) using a 5× detection objective, 5× illumination optics, and laser excitation at 647nm. All images were captured at 1.1× zoom, with z-stack intervals set at 4.8 μm and an exposure time of 59.92 ms. Three-dimensional-rendered images were visualized with Zen 3.1 (blue edition, Zeiss). The brightness and contrast of the 3D-rendered images were manually adjusted to aid visual clarity.

**In situ hybridization:** Embryonic heads (E15.5) and extracted brains (P10, P21) were rapidly frozen on dry-ice in plastic molds filled with optimal cutting medium (O.C.T; Sakura, Torrance, CA, USA), and cryosectioned at 16-µm thickness on a Leica CM1860 cryostat microtome. Coronal sections containing the PVN were collected serially on SuperFrost^+^ glass slides (ThermoFisher, Vienna, Austria), air-dried for 20 min, and stored at −80 °C until processing. Tissue sections were immersion fixed in 4% paraformaldehyde (PFA) in 0.05M phosphate-buffered saline (PBS, pH 7.4) at 4 °C for 20 min, rinsed in PBS, and subsequently dehydrated in an ascending gradient of ethanol (25%, 50%, 75%, and 100%, 5 min each). In situ hybridization was multiplexed according to the hybridization chain reaction (HCR) v3.0 protocol for ‘generic sample on slide’ with probe sets combining *Slc32a1*, *Crh*, *Trh*, *Oxt*, *Cnr1*, *Dagla*, *Daglb*, and *Mgll* (all from Molecular Instruments, Los Angeles, CA, USA). Sections were counterstained with the nuclear dye Hoechst 33,342 (Sigma Aldrich, Vienna, Austria) and mounted with Entellan (Merck, Vienna, Austria). Sections with two or more colors from appropriately labeled hairpin combinations were imaged on an LSM880 confocal microscope (Zeiss; pinhole set to 1 airy unit and minimal laser power [<5% per channel]; 20×/0.8 NA objective for survey images and Plan-Apochromat 63×/1.4 N.A. objective for high-resolution images), processed with the ZEN 3.0 SR software (Zeiss), and compiled as multi-panel images in CorelDRAW 2019 (Corel Corp., Austin, TX, USA). Representative images of *n* = 2−3 mice were incorporated in the figure panels.

**Immunohistochemistry.** Whole heads of mouse fetuses (E15.5) were immersion fixed overnight with 4% PFA in 0.05 M PBS at 4 °C before cryoprotection in 30% sucrose in 0.05 M PBS. P21 mice were transcardially perfused with 4% PFA in 0.1 M phosphate buffer (PB, pH 7.4), with their brains removed and post-fixed at 4 °C overnight, before cryoprotection in 30% sucrose in 0.05 M PBS. Tissues were rapidly frozen and cryosectioned on a Leica CM1860 cryostat at either 20 µm (E15.5) and collected on SuperFrost^+^ glass slides (Thermo Fisher, Austria) or 50 µm for free-floating labeling (P21). Next, sections were incubated with a blocking solution containing 5% normal donkey serum (NDS, Jackson ImmunoResearch, West Grove, PA, USA), 2% bovine serum albumin (BSA, Sigma), and 0.2% Triton X-100 (Sigma Aldrich, Austria) in PBS at room temperature for 1 h to block non-specific binding. Tissues were subsequently exposed to a combination of primary antibodies (guinea pig anti-CB_1_R [Af530], 1:500, Nittobo Medical; goat anti-DAGLα [Af1080], 1:500, Nittobo Medical; and rabbit anti-oxytocin [AB911], 1:1000; Merck Millipore, Vienna, Austria) diluted in 2% NDS, 0.1% BSA, and 0.2% Triton X-100 in PBS at 4 °C for 72 h. After extensive rinsing in PBS, appropriate combinations of secondary IgGs conjugated with carbocyanine (Cy)2, 3, or 5 (raised in donkey, 1:300, Jackson ImmunoResearch) were applied at 22–24 °C for 2 h. Sections were counterstained with the nuclear dye Hoechst 33,342 (Sigma Aldrich) to visualize nuclei. After extensive washing in PBS, sections were rinsed in distilled water, air-dried, and cover slipped with Entellan (in xylene, Sigma-Aldrich). Representative images of *n* = 2–3 mice were incorporated in the figure panels.

## Figures and Tables

**Figure 1 cells-14-00788-f001:**
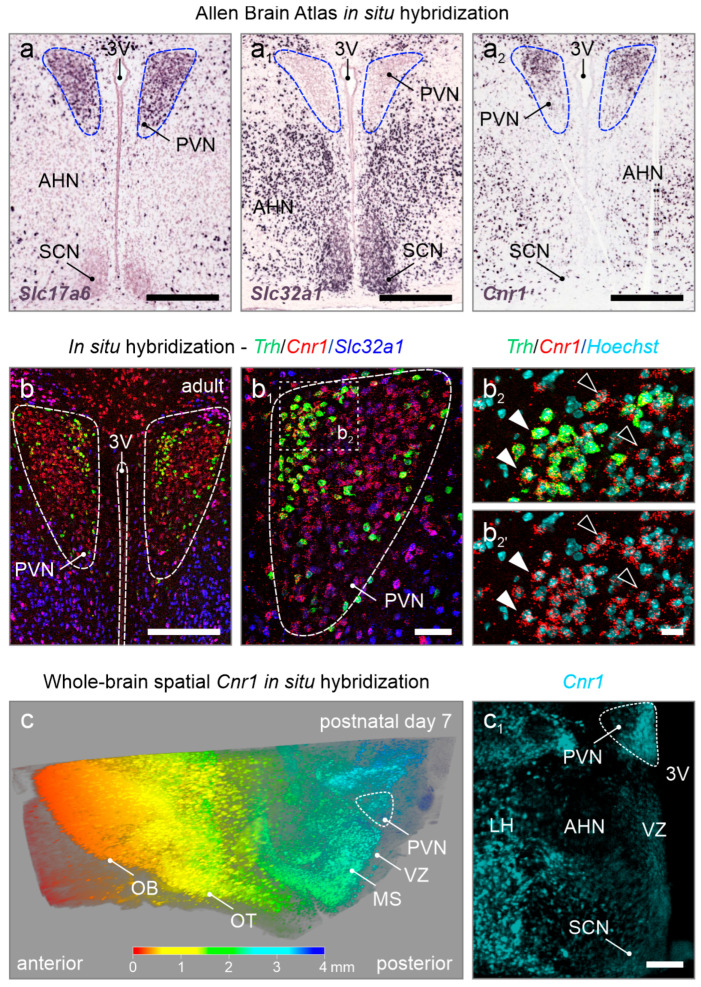
*Cnr1* mRNA expression in the adult mouse PVN. (**a**–**a_2_**) Open-source in situ hybridization data from the Allen Brain Atlas (https://portal.brain-map.org/, accessed on 22 May 2025) in the adult mouse PVN reveals glutamatergic (*Slc17a6^+^*) but not GABAergic (*Slc32a1^+^*) neurons, along with the accumulation of *Cnr1* mRNA. (**b**–**b_2’_**) Multiplexed in situ hybridization confirmed *Cnr1* mRNA expression in the PVN at levels more pronounced than in neighboring regions, with *Trh*^+^ neurons being particularly labeled (closed vs. open arrowheads). Hoechst 33,342 was used as a nuclear counterstain. (**c**,**c_1_**) Three-dimensional reconstruction of a cleared postnatal day 7 hemisphere processed for *Cnr1* mRNA detection. Note the high amount of *Cnr1* expression throughout the anterior–posterior axis (rainbow color-coded), with the PVN being highlighted in the hypothalamic area (**c_1_**). Abbreviations: 3V, third ventricle; AHN, anterior hypothalamic nucleus; LH, lateral hypothalamus; MS, medial septum; OB, olfactory bulb; OT, olfactory tract; PVN, paraventricular nucleus; SCN, suprachiasmatic nucleus; VZ, ventricular zone. Scale bars = 350 µm (**a**–**a_2_**,**c_1_**), 200 µm (**b**), 50 µm (**b_1_**), and 20 µm (**b_2_**).

**Figure 2 cells-14-00788-f002:**
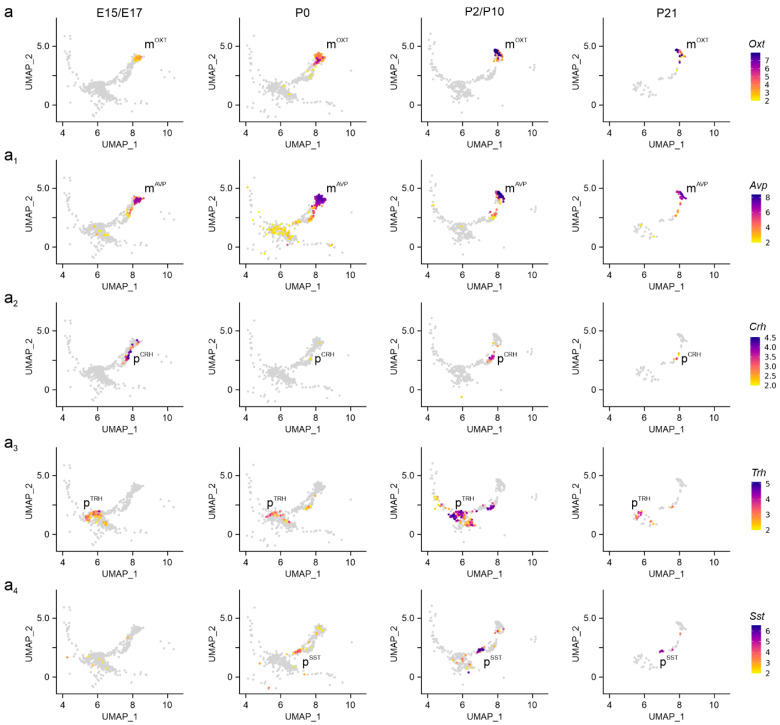
Neuropeptide identifiers of the PVN. (**a**–**a_4_**) Multidimensional clustering based on neuropeptides revealed discrete magnocellular (m^OXT^, m^AVP^) and parvocellular cells, the latter containing *Crh*, *Trh*, and *Sst* (p^CRH^, p^TRH^, p^SST^) across successive developmental stages in the PVN. Relative expression was color-coded for each gene analyzed (to the right).

**Figure 3 cells-14-00788-f003:**
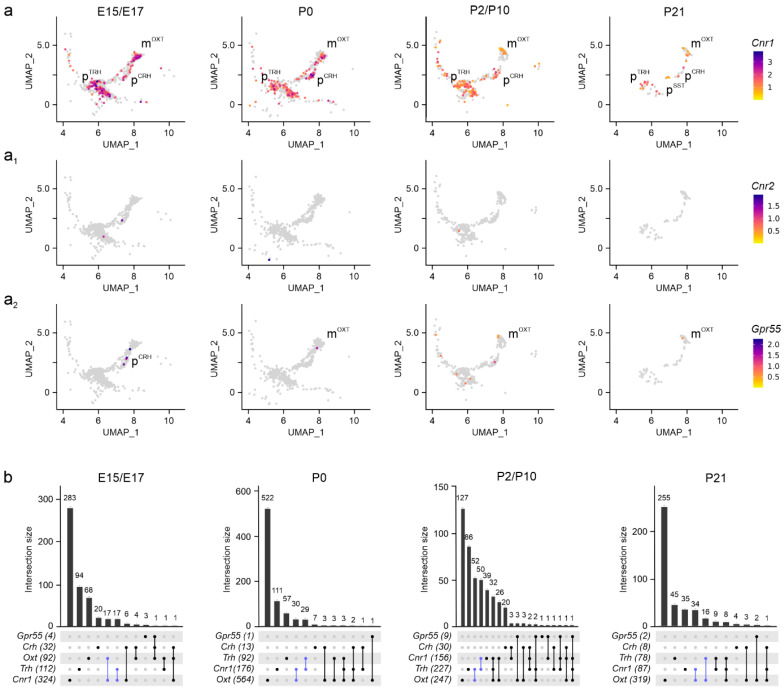
Cannabinoid receptors in the PVN. (**a**–**a_2_**) UMAP representation of cannabinoid receptors identified *Cnr1* as the dominant receptor in both magnocellular and parvocellular cells at all developmental stages analyzed, with minimal contributions from *Cnr2* (**a_1_**) and *Gpr55* (**a_2_**). Relative expression is color-coded for each gene analyzed (to the right). (**b**) *UpSet* plots showing gene enrichment and co-expression alongside brain development. Note the particular enrichment in m^OXT^ and p^TRH^ clusters (in blue) compared to p^CRH^.

**Figure 4 cells-14-00788-f004:**
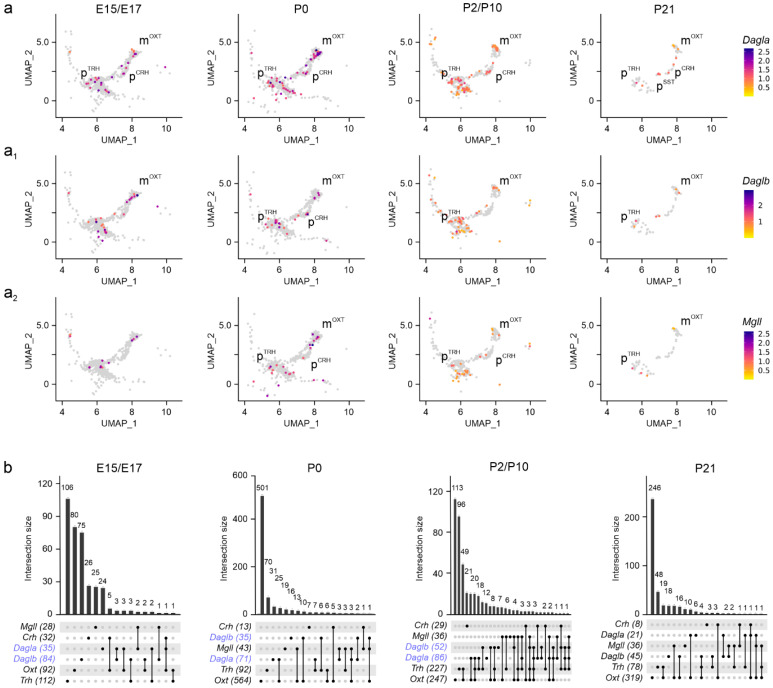
*Dagla/b* and *Mgll* expression in the PVN. (**a**–**a_2_**) UMAP representation in developing neurons showed *Dagla* < *Daglb* until birth (**b**, in blue), and *Mgll* at all time points. Relative expression is color-coded (to the right) for each gene analyzed. (**b**) *UpSet* plots showing gene enrichment and co-expression alongside brain development for the genes analyzed.

**Figure 5 cells-14-00788-f005:**
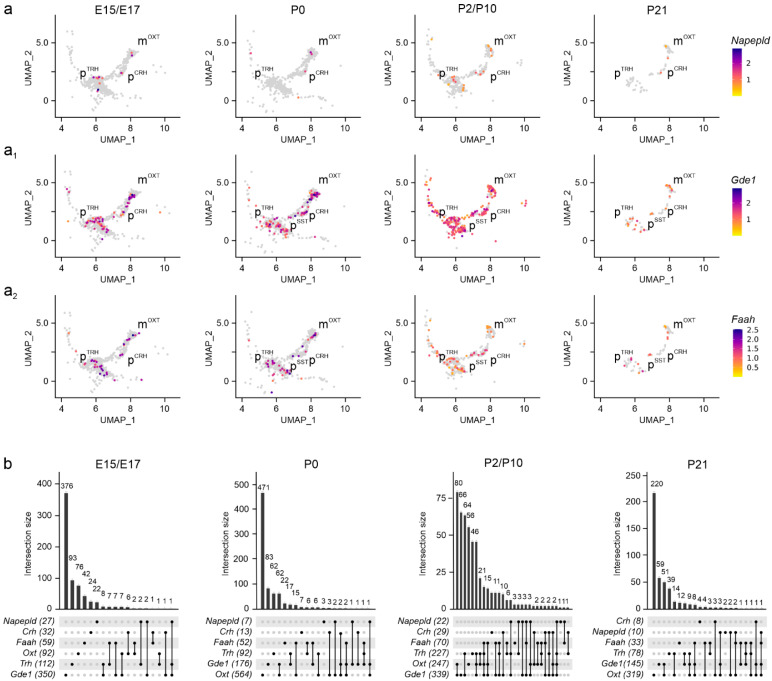
*Napepld*, *Gde1*, and *Faah* expression in the PVN. (**a**–**a_2_**) UMAP representation of *Napepld*, *Gde1*, and *Faah* expression in neurons populating the PVN. Relative expression is color-coded (to the right) for each gene analyzed. (**b**) *UpSet* plots showing gene enrichment and co-expression alongside brain development for the genes analyzed.

**Figure 6 cells-14-00788-f006:**
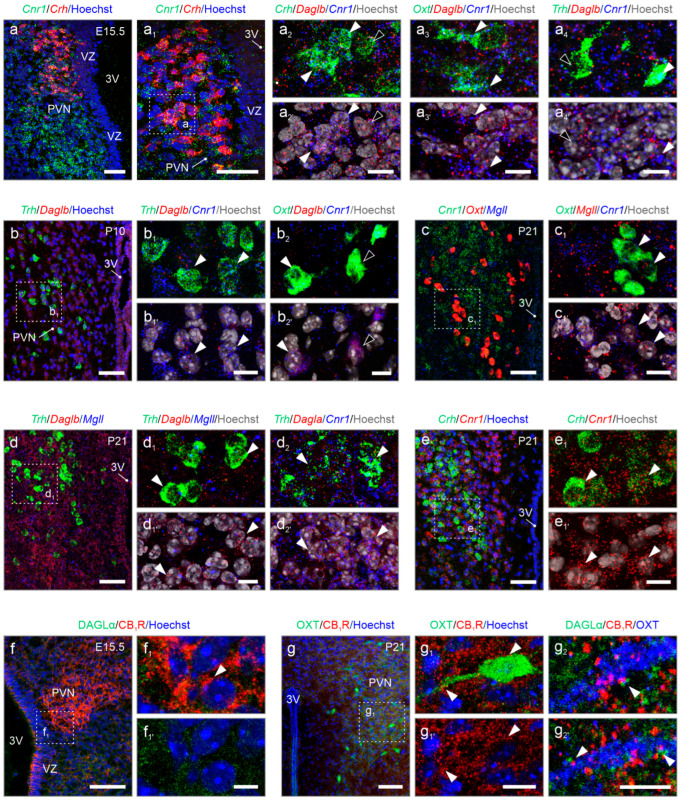
Cell-type-specific localization of *Dagla*, *Daglb*, *Magll*, and Cnr1 by in situ hybridization. (**a**–**a_4’_**) Already at E15.5, distinct neuronal populations express either *Cnr1* or *Daglb*, or both. Arrowheads indicate co-localization, while open arrowheads denote the absence. (**b**) At postnatal day 10 (P10), both *Cnr1* and *Daglb* expression were maintained. (**c**–**e_1’_**) At P21, PVN neurons continue to express *Cnr1*, *Daglb*, and *Mgll*, along with noticeable levels of *Dagla* (arrowheads, **d_2_**). (**f**–**g_2’_**) Immunohistochemistry reveals the focal accumulation of CB_1_Rs in the PVN at E15.5, which remains unchanged until P21 (arrowheads, **f_1_** vs. **g_1_**). DAGLα dominates at P21, when it is frequently apposed to CB_1_Rs in the somata and processes of, e.g., m^OXT^ neurons (arrowheads, **g_2_**). Tissues were counterstained with Hoechst 33,342 to visualize nuclei. Scale bars = 50 µm (**a**,**a_1_**,**b**,**c**,**d**,**e**,**f**,**g**), 20 µm (**g_1_**), 10 µm (**a_2_**,**a_3_**,**a_4_**,**b_1_**,**b_2_**,**c_1_**,**d_1_**,**d_2_**,**e_1_**), and 5 µm (**f_1_**,**g_2_**).

**Figure 7 cells-14-00788-f007:**
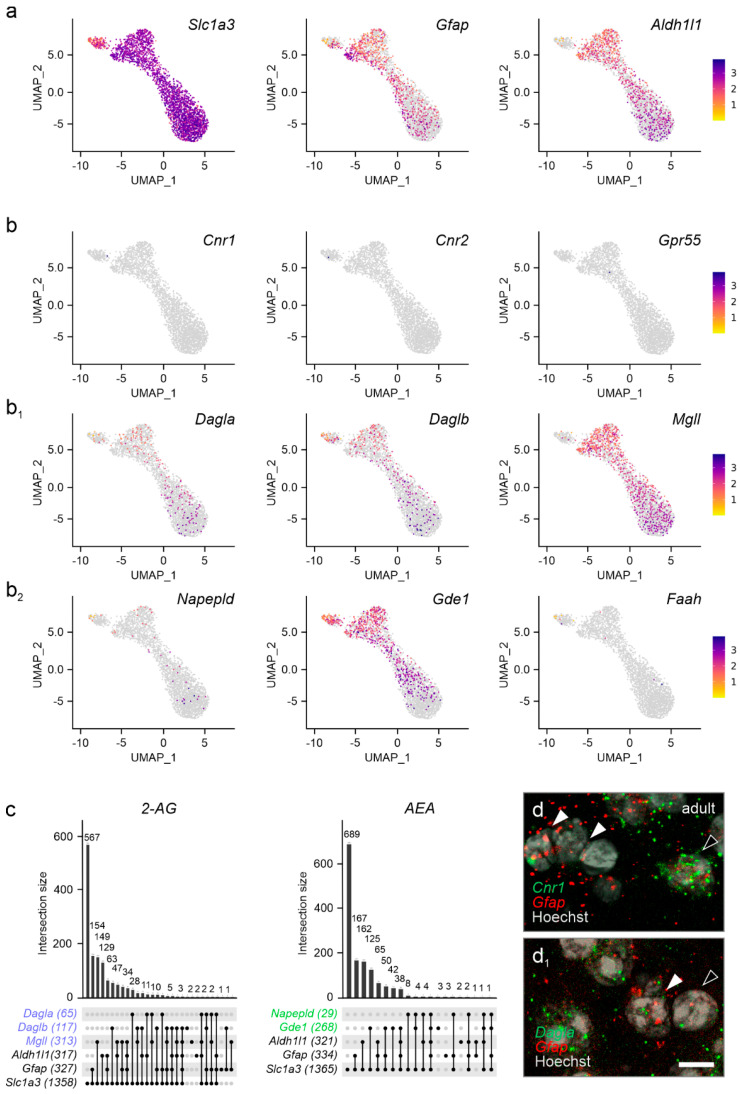
Cannabinoid receptors and enzymes in astrocytes of the adult PVN. (**a**) Gene set used to identify astrocytes. (**b**–**b_2_**) Cannabinoid receptors (**b**), but not the machinery to control the bioavailability of 2-AG, are lacking in astrocytes. Relative expression is color-coded for each gene analyzed (to the right). (**c**) *UpSet* plots for gene enrichment in PVN astrocytes. Note the higher reliance on 2-AG signaling as compared to AEA (blue vs. green). (**d**) In situ hybridization confirms the lack of *Cnr1* in *Gfap*^+^ astrocytes (arrowheads indicate *Gfap^+^/Cnr1*^−^ astrocytes vs. open arrowheads denoting *Gfap*^−^/*Cnr1*^+^ cells, presumed neurons). (**d_1_**) *Gfap*^+^ astrocytes contain *Dagla* mRNA, even if at low abundance (arrowheads). Scale bar = 5 µm (**d_1_**).

## Data Availability

The snRNA-seq data generated in this study have been deposited in the NCBI Gene Expression Omnibus database under accession code (GSE132730). All primary data were made publicly available at https://github.com/EugOT/eCB-hypothalamus-development (accessed on 22 May 2025) and deposited to Figshare.com with DOI: 10.6084/m9.figshare.28816100. All other relevant data were included in the figures.

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
