# Peer review of "Molecular Fingerprint of Endocannabinoid Signaling in the Developing Paraventricular Nucleus of the Hypothalamus as Revealed by Single-Cell RNA-Seq and In Situ Hybridization"

_cells, 2025, doi:10.3390/cells14110788_

Round 1

Reviewer 1 Report

Comments and Suggestions for Authors

This is an interesting data set that examines the early developmental time course of the expression of components of endocannabinoid signaling in the PVN.  In general, the findings replicate other data and support the role of CB1R/2-AG signaling in early stages of brain development.

A few suggestions for improvement

  1. The structure of the paper has some unconventional features. First, figure 1 is part of the introduction rather than the results.  This should be changed.  And, second, there are instances in the results in which conclusions are made about the data; this should be done in the discussion section.  An example of the second issue is seen on line 199, where the data are interpreted to mean that 2-AG is superior to AEA in the PVN.
  2. For the immunohistochemistry and in situ hybridization studies, some form of quantification should be employed. In addition, are these representative images, or the only images that were made?
  3. It would be easier to understand the data quickly if the UpSet plots used the same gene orders
  4. Line 299: although 2-AG concentrations in brain tissue are much higher than AEA concentrations, Larry Parsons showed many years ago that synaptic concentrations (determined using microdialysis) of the two are very similar. Much of the tissue 2-AG is likely a storage form of arachidonic acid.
  5. Line 321: what is meant by CB1R-independent 2-AG release? Isn’t all 2-AG release CB1R-independent?
  6. Supplemental figure 1: There is no color code on the figure.
Comments on the Quality of English Language
  1. Suggested wording edits:
    1. Line 20: neuromodulator should be neuromodulation
    2. Line 24: arachydonoylglycerol should be arachidonoylglycerol
    3. Line 89: towards should be until
    4. Line 94: lastly should be last
    5. Line 143: phrase starting “even if in a …” is not clear
    6. Line 200: reminisced should be is reminiscent of

Author Response

Dear Reviewer,

Thank you for your constructive criticisms and suggestions for the improvement of our manuscript. We have addressed your queries below and marked all changes in the manuscript in blue.

Comment 1: The structure of the paper has some unconventional features. First, figure 1 is part of the introduction rather than the results.  This should be changed.  And, second, there are instances in the results in which conclusions are made about the data; this should be done in the discussion section.  An example of the second issue is seen on line 199, where the data are interpreted to mean that 2-AG is superior to AEA in the PVN.

Response 1: Thank you for pointing out our unconventional features. To adhere to a more classical construction, we moved Fig. 1 and its description to the results section. In addition, interpretations of data have been toned down in the results section.

Comment 2: For the immunohistochemistry and in situ hybridization studies, some form of quantification should be employed. In addition, are these representative images, or the only images that were made?

Response 2: We have added a semi-quantitative analysis of the cell-specific mRNA content of the 2-AG signaling cassette throughout development in Supporting Table 1. The images presented in Figure 6 are representative pictures chosen from a selection of images made from n = 2-3 animals per developmental time point. 

Comment 3: It would be easier to understand the data quickly if the UpSet plots used the same gene orders

Response 3: The UpSet plots were constructed to visualize the intersection of set sizes through decremental cell numbers on the x-axis, as well as for the genes labelled below the plots. This allows for the only correct interpretation of which gene combinations from our analyzed pool are occurring at which frequency. Although presenting the UpSet plots with same gene orders could potentially improve its readability for readers interested in endocannabinoids, it would render the scientific meaning of gene expression intersection moot, which we wish to avoid at all cost.

Comment 4: Line 299: although 2-AG concentrations in brain tissue are much higher than AEA concentrations, Larry Parsons showed many years ago that synaptic concentrations (determined using microdialysis) of the two are very similar. Much of the tissue 2-AG is likely a storage form of arachidonic acid.

Response 4: Thank you for bringing our attention to this, we have added a disclaimer in this section.

Comment 5: Line 321: what is meant by CB1R-independent 2-AG release? Isn’t all 2-AG release CB1R-independent?

Response 5: “CB1R-independent” has been removed.

Comment 6: Supplemental figure 1: There is no color code on the figure.

Response 6: The color code has been added.

Comment 7: Suggested wording edits.

Line 20: neuromodulator should be neuromodulation

Line 24: arachydonoylglycerol should be arachidonoylglycerol

Line 89: towards should be until

Line 94: lastly should be last

Line 143: phrase starting “even if in a …” is not clear

Line 200: reminisced should be is reminiscent of

Response 7: We have incorporated the above suggestions.

Reviewer 2 Report

Comments and Suggestions for Authors

In their study, based on single cell RNA sequencing and in situ hybridization, Tretiakov et al. show among others that CB1 receptors (but not CB2 and GPR55 receptors) are expressed in neurones of the PVN and that none of the latter three receptors but enzymes involved in synthesis and degradation of the endocannabinoid 2-arachidonoyl glycerole are expressed in astrocytes of this hypothalamic brain region. Data are sound but their presentation is sometimes complicated. Note that this reviewer has published numerous papers in the field of cannabinoids and PVN, yet has difficulties to follow all sentences. Some recommendations to overcome the problem will be given below.

A list explaining all abbreviations is mandatory.

Some results are given in the Summary but only an extreme expert will be able to understand what the authors want to say (l. 27-29). I am wondering whether the authors should explain that Dagla/Daglb and Mgll are involved in the synthesis and degradation of 2-AG, respectively, whereas Napepld and Faah serve the synthesis and degradation of AEA, respectively.

l. 85-104: Well, authors frequently use the last paragraph of the Introduction to summarize the results. I would suggest that the present authors should rather explain the design of the study, e.g., which parameters they studied, which tissues they used, that they examined the pre- and early postnatal development of some parameters in neurones and that they correlated RNA expression and protein distribution.

l. 86: I have some difficulties with "metabolism". Strictly speaking, this term refers to degradation but not to synthesis of a biomolecule.

l. 109: I am not quite sure but please check whether "Supporting..." is correct. To the best of my knowledge, it should read "Supplementary..." instead.

l. 111: "believe" instead of "caution"?

l. 112-113: "for this reason, Avp was excluded" instead of  "for which Avp itself"?

l. 132, Figure 2: Is it correct to combine data obtained on P2 and P10?

l. 133: Should it not read a-a4?

l. 150: "whether" is not the correct word here.

l. 175 and Figure 4: Is there an explanation that Oxt and Trh are combined?

l. 179: The role of Gde1, compared to that of Napepld, should be explained.

l. 200: "was reminiscent of" instead of "reminisced"?

l. 231 and Figure 7: The AEA panel would be even more interesting if Faah were added and explained that Gde1 is unspecific.

l. 270, 303, 385 and 395: Abbreviations should be explained on first mention, including BNST, HPA, HCR and PFA, respectively.

l. 316: "also other cannabinoid receptors" instead of "also others"?

l. 334-412 (Methods section): Hoechst, UMAP and UpSet should be explained in a manner that a scientist not working with these tools can understand what they really mean.

It is unclear to me whether the data shown in the figures have been generated once only or have been generated several times and the best result is shown in the manuscript only.

Author Response

Dear Reviewer, thank you for your constructive criticisms to improve the flow and readability of our manuscript, especially for non-experts. As such, we have addressed all your queries and incorporated them into this revised version (all changes are in blue).

Comment 1. A list explaining all abbreviations is mandatory.

Response 1. We have added a list of abbreviations to the Supplementary Information.

Comment 2. Some results are given in the Summary but only an extreme expert will be able to understand what the authors want to say (l. 27-29). I am wondering whether the authors should explain that Dagla/Daglb and Mgll are involved in the synthesis and degradation of 2-AG, respectively, whereas Napepld and Faah serve the synthesis and degradation of AEA, respectively.

Response 2. The Abstract has been updated to be more explanatory.

Comment 3. l. 85-104: Well, authors frequently use the last paragraph of the Introduction to summarize the results. I would suggest that the present authors should rather explain the design of the study, e.g., which parameters they studied, which tissues they used, that they examined the pre- and early postnatal development of some parameters in neurones and that they correlated RNA expression and protein distribution.

Response 3. We have updated the last paragraph of the introduction with the proposed suggestions.

Comment 4. l. 86: I have some difficulties with "metabolism". Strictly speaking, this term refers to degradation but not to synthesis of a biomolecule.

Response 4. The word metabolism has been changed to “synthesis and degradation”. However, please note that the Oxford Dictionary states: “the chemical processes that occur within a living organism in order to maintain life. … Two kinds of metabolism are often distinguished: constructive metabolism (= anabolism), the synthesis of the proteins, carbohydrates, and fats which form tissue and store energy, and destructive metabolism (= catabolism), the breakdown of complex substances and the consequent production of energy and waste matter." Thus, we are steadfast in our view that the use of “metabolism” was correct.

Comment 5. l. 109: I am not quite sure but please check whether "Supporting..." is correct. To the best of my knowledge, it should read "Supplementary..." instead.

Response 5. Indeed, we erroneously have used “Supporting” and have changed it to “Supplementary”.

Comment 6. l. 111: "believe" instead of "caution"?

Response 6. In this case, “caution” would be more fitting as we would like to make the reader aware that the sequence similarity is limiting the preciseness of our analysis.

Comment 7. l. 112-113: "for this reason, Avp was excluded" instead of  "for which Avp itself"?

Response 7. We have changed the sentence.

Comment 8. l. 132, Figure 2: Is it correct to combine data obtained on P2 and P10?

Response 8. As P2 and P10 are fitting between the two major developmental milestones birth and puberty it is considered valid to combine these two time points.

Comment 9. l. 133: Should it not read a-a4?

Response 9. Correct, we have changed this.

Comment 10. l. 150: "whether" is not the correct word here.

Response 10. This has been corrected now.

Comment 11. l. 175 and Figure 4: Is there an explanation that Oxt and Trh are combined?

Response 11. In our single cell RNA database, we found a small population of neurons that co-express Oxt and Trh, which are indeed visualized in the UpSet plots.

Comment 12. l. 179: The role of Gde1, compared to that of Napepld, should be explained.

Response 12. This has been added .

Comment 13. l. 200: "was reminiscent of" instead of "reminisced"?

Response 13. This has been changed.

Comment 14. l. 231 and Figure 7: The AEA panel would be even more interesting if Faah were added and explained that Gde1 is unspecific.

Response 14. As there was no co-localization detected between Faah and the three main astrocyte markers used (Fig. 7a), the algorithm did not include this gene in the AEA panel. The lack of specificity of Gde1 has been emphasized in the discussion.

Comment 15. l. 270, 303, 385 and 395: Abbreviations should be explained on first mention, including BNST, HPA, HCR and PFA, respectively.

Response 15. This has been checked and changed.

Comment 16. l. 316: "also other cannabinoid receptors" instead of "also others"?

Response 16. This has been changed accordingly.

Comment 17. l. 334-412 (Methods section): Hoechst, UMAP and UpSet should be explained in a manner that a scientist not working with these tools can understand what they really mean.

Response 17. These have been expanded.

Comment 18. It is unclear to me whether the data shown in the figures have been generated once only or have been generated several times and the best result is shown in the manuscript only.

Response 18. The images presented for in situ hybridization and immunohistochemistry are representative pictures chosen from a selection of pictures made from n = 2-3 animals per developmental time point. 

Round 2

Reviewer 2 Report

Comments and Suggestions for Authors

The manuscript by Tretiakov et al. has been revised by the authors and is very close to final acceptance (at least from my point of view). I have three points.

  1. Is it correct to say "opposing" in l. 102 but "apposed" in l. 234?
  2. The authors have explained in their rebuttal letter that the figures are representative examples of several slides obtained from 2 to 3 animals. I am wondering whether this piece of information should be included in the Methods section.
  3. The list of abbreviations should be part of the main manuscript. By the way, I did not receive a supplementary file in which abbreviations are explained.

Author Response

Dear Reviewer,

Thank you for your help in improving this manuscript even further. We have addressed your comments below.

Comment 1: Is it correct to say "opposing" in l. 102 but "apposed" in l. 234?

Response 1: We have corrected the one instance of "oppose" to "appose".

Comment 2: The authors have explained in their rebuttal letter that the figures are representative examples of several slides obtained from 2 to 3 animals. I am wondering whether this piece of information should be included in the Methods section.

Response 2: We have added this statement to the material and methods.

Comment 3: The list of abbreviations should be part of the main manuscript. By the way, I did not receive a supplementary file in which abbreviations are explained.

Response 3: The list of abbreviations has now been included in the main file (before the references).